# Adjuvant HPV Vaccination to Prevent Recurrent Cervical Dysplasia after Surgical Treatment: A Meta-Analysis

**DOI:** 10.3390/vaccines9050410

**Published:** 2021-04-21

**Authors:** Violante Di Donato, Giuseppe Caruso, Marco Petrillo, Evangelos Kontopantelis, Innocenza Palaia, Giorgia Perniola, Francesco Plotti, Roberto Angioli, Ludovico Muzii, Pierluigi Benedetti Panici, Giorgio Bogani

**Affiliations:** 1Department of Maternal and Child Health and Urological Sciences, Sapienza University of Rome, Policlinico Umberto I, 00161 Rome, Italy; violante.didonato@uniroma1.it (V.D.D.); innocenza.palaia@uniroma1.it (I.P.); giorgia.perniola@uniroma1.it (G.P.); ludovico.muzii@uniroma1.it (L.M.); pierluigi.benedettipanici@uniroma1.it (P.B.P.); 2Gynecologic and Obstetric Unit, Department of Medical, Surgical and Experimental Sciences, University of Sassari, 07100 Sassari, Italy; marco.petrillo@gmail.com; 3PhD School in Biomedical Sciences, University of Sassari, 07100 Sassari, Italy; 4Division of Informatics, Imaging and Data Science, University of Manchester, Manchester M13 9PL, UK; e.kontopantelis@manchester.ac.uk; 5Department of Obstetrics and Gynecology, Campus Bio-Medico University of Rome, 00128 Rome, Italy; f.plotti@unicampus.it (F.P.); r.angioli@unicampus.it (R.A.); 6Department of Gynecologic Oncology, IRCCS National Cancer Institute, 20133 Milan, Italy; giorgio.bogani@istitutotumori.mi.it

**Keywords:** cervical intraepithelial neoplasia, cervical dysplasia, conization, human papillomavirus, HPV, LEEP, vaccination

## Abstract

Objective: The aim of this meta-analysis was to discuss evidence supporting the efficacy of adjuvant human papillomavirus (HPV) vaccination in reducing the risk of recurrent cervical intraepithelial neoplasia (CIN) 2 or greater after surgical treatment. Methods: A systematic literature search was performed for studies reporting the impact of HPV vaccination on reducing the risk of recurrence of CIN 2+ after surgical excision. Results were reported as mean differences or pooled odds ratios (OR) with 95% confidence intervals (95% CI). Results: Eleven studies met the inclusion criteria and were selected for analysis. In total, 21,310 patients were included: 4039 (19%) received peri-operational adjuvant HPV vaccination while 17,271 (81%) received surgery alone. The recurrence of CIN 2+ after treatment was significantly lower in the vaccinated compared with the unvaccinated group (OR 0.35; 95% CI 0.21–0.56; *p* < 0.0001). The recurrence of CIN 1+ after treatment was significantly lower in the vaccinated compared with the unvaccinated group (OR 0.51; 95% CI 0.31–0.83; *p* = 0.006). A non-significant trend of reduction rate of HPV persistence was observed in the vaccinated compared with the unvaccinated cohorts (OR was 0.84; 95% CI 0.61–1.15; *p* = 0.28). Conclusions: HPV vaccination, in adjuvant setting, is associated with a reduced risk of recurrent CIN 1+ and CIN 2+ after surgical treatment.

## 1. Introduction

Human papillomavirus (HPV) infections are responsible for the majority of cervical cancer cases [1,2]. In 2018, HPV was responsible for 690,000 new cancer cases worldwide [3]. Globally, cervical cancer represented about 80% (*n* = 570,000 cases) of the HPV-attributable cancer burden [3]. HPV types 16 and 18 were estimated to account for nearly 90% of HPV-related cervical cancers [3]. In 2017, 7085 HPV-attributable cancer deaths occurred in the US with a total of 15,454 years of potential life lost (YPLL). The highest YPLL was associated with cervical cancer (65% of all YPLL) [4].

Despite the increasing introduction of HPV vaccines and screening strategies, cervical cancer still remains the first most common gynecological cancer worldwide [1,5]. Currently, over one hundred countries around the world, accounting for 30–50% of the target population, have included HPV vaccination into their routine prevention programs [6,7,8]. The HPV vaccination is the most cost-effective public health measure against cervical cancer and it represents the key pillar to prevent invasive cervical cancer [9]. Three types of vaccines are available at present (bivalent, quadrivalent, and nonavalent) and they all target at least the two most oncogenic virus genotypes (HPV 16, 18), which are responsible for over 70% of cervical cancers [10,11]. Routine prophylactic vaccination of women should be recommended at 11–12 years to ensure its effectiveness before sexual activity [12].

The main goal of cervical screening programs is to detect early and treat pre-cancerous lesions (i.e., high-grade cervical intraepithelial neoplasia (CIN 2+) or high-grade squamous intraepithelial lesions (HSIL)). Preneoplastic cervical lesions are generally treated with excisional surgery, most commonly using loop electrosurgical excision procedure (LEEP) or conization (either with laser or cold knife) [13]. Nevertheless, the overall risk of recurrence after surgical treatment for CIN 2+ is around 10–14%, with 6% and 16.5% of patients having recurrent CIN 3+ and CIN 2+ at 5 years, respectively [14].

Enormous attempts are being made to obtain the first therapeutic HPV vaccine [15,16,17]. In the meantime, interesting data suggest that the administration of prophylactic HPV vaccine, either shortly before or after surgical treatment for CIN 2+, might reduce the risk of recurrence [18,19,20]. The rationale behind the efficacy of HPV vaccine as an adjuvant therapy remains unclear. HPV vaccines elicit the development of neutralizing antibodies against HPV-like particles, eventually preventing virus particles from entering into host cells. These vaccines should not be effective in clearing preexistent infections as viral antigens are not expressed on the surface of infected cells and so cannot be targeted by antibodies after vaccination [21,22]. Several hypotheses have been proposed so far to explain how even women already infected may benefit from HPV vaccination. The simpler one is that HPV vaccination may provide cross protection against other HPV strains to which patients have not been previously exposed, thus preventing new infections [23,24]. However, the strong association between the HPV type-specific persistence (mainly genotypes HPV 16 and 18) after treatment for cervical dysplasia and the recurrent disease would suggest a different mechanism than a new infection [25,26]. Therefore, it has been also hypothesized that the surgical treatment may reduce the local inflammatory response, induce a higher magnitude and more durable local cellular immunity, and restore an HPV-naïve microenvironment, in which the HPV vaccine could theoretically be efficacious in preventing persistent and recurrent HPV infections [27,28,29]. Nevertheless, there is currently insufficient evidence to recommend adjuvant HPV vaccination along with the surgical excision of cervical dysplasia. The present systematic review and meta-analysis summarizes the currently available data on the efficacy of perioperative prophylactic HPV vaccination in reducing the risk of recurrence of CIN 2+ after surgery.

## 2. Materials and Methods

### 2.1. Search Strategy

A systematic literature search up to 12 February 2021 was performed for all English-language publications reporting risk of recurrence of cervical intraepithelial neoplasia after surgical treatment in patients receiving prophylactic HPV vaccination. MEDLINE, Embase, PubMed, Cochrane databases, and clinicaltrials.gov were searched using a Boolean search algorithm for articles published up to February 2021. The process of evidence acquisition combined the following search terms: “cervical cancer”, “cervical disease”, “cervical intraepithelial neoplasia”, “human papillomavirus”, “vaccine”, and “vaccination” (Appendix A).

Additional screening was performed of the reference lists from the relevant literature. Article abstracts and, where appropriate, full text of articles and cross-referenced studies identified from retrieved articles were screened for pertinent information. All duplicate records were removed. The overall search strategy was performed using PRISMA (Preferred Reporting Items for Systematic Reviews and Meta-Analyses) guidelines [30].

### 2.2. Selection of Studies and Methodologic Quality Assessment

The study selection was done independently by two authors (G.C., V.D.D). The publications were evaluated dependent on predefined inclusion and exclusion criteria. The inclusion criteria were as follows: (1) randomized controlled, prospective or retrospective observational studies; (2) patients undergoing surgery for HPV-related cervical dysplasia; (3) prophylactic HPV vaccination (either shortly before or after surgery) versus no vaccination; (4) histologically confirmed CIN 2+ recurrence. The exclusion criteria were as follows: (1) case reports, case studies, editorials, review articles and conference abstracts; (2) studies testing newly developed HPV vaccines that have not received FDA approval; (3) studies in which participants had invasive disease, immunodeficiency or autoimmune conditions, received systemic corticosteroids, or were pregnant.

Data extraction from each included study was done on the basis of study characteristics and predefined outcome variables. The following items were extracted from each study: journal and year of publication, study design, study endpoint, study population (age and number of patients), type of treatment (conization, LEEP, cryotherapy), type of HPV vaccine (bivalent, quadrivalent, nonavalent), time of vaccination (before or after surgery), duration of follow-up, number of CIN recurrences. Discrepancies of opinion between the reviewers were resolved by consensus.

The methodological quality of each study was assessed according to how patients were allocated to the arms of the study, the concealment of allocation procedures, blinding, and data loss due to attrition. The studies were then classified qualitatively according to the guidelines published in the Cochrane Handbook for Systematic Reviews of Interventions v.5.1.0 [31]. Based on the quality assessment criteria, each study was rated and assigned to one of the three following quality categories: A, if all quality criteria were adequately met, the study was deemed to have a low risk of bias; B, if one or more of the quality criteria was only partially met or was unclear, the study was deemed to have a moderate risk of bias; or C, if one or more of the criteria was not met or not included, the study was deemed to have a high risk of bias. Differences were resolved by discussion among the authors.

A further quality score was used to evaluate whether any studies were useful for the purpose of our analysis [32,33,34,35]. The quality score was calculated by summing the singular score assigned to each of following variables: type of study (prospective = 2; retrospective or population-based = 1), total number of patients per study (<100 = 1; 100–150 = 2; >150 = 3), volume of the center (number of patients per year of study: <20 = 1; 20–50 = 2; >50 = 3). All studies with a quality score of <4 were excluded. Ethical approval was not required as data from previous published studies were retrieved and analyzed.

### 2.3. Outcomes

Outcomes that are particularly concerning for patients in this context were selected:(1)Primary outcomes
1.1CIN 2+ recurrence: recurrences of CIN 2+ (irrespective of HPV type and HPV 16/18-related) have been extrapolated from the studies.1.2CIN 1+ recurrence: recurrences of CIN 1+ (irrespective of HPV type and HPV 16/18-related) have been extrapolated from the studies.(2)Secondary outcomes
2.1HPV persistence: the detection of positive HPV test at the first clinical follow-up after surgical treatment.2.2CIN 3 recurrence: recurrences of CIN 3 (irrespective of HPV type and HPV 16/18-related) have been extrapolated from the studies.2.3Persistence of abnormal cervical cytology: the detection of abnormal cervical cytology at the first clinical follow-up after surgical treatment.

### 2.4. Statistical Analysis

The data were analyzed using RevMan software (Review Manager version 5.3, the Cochrane Collaboration). Dichotomous outcomes from each study were expressed as an odds ratio (OR) with a 95% confidence interval (CI). Heterogeneity between studies was reported with the I^2^ statistic. A “hybrid” Mantel–Haenszel random-effects model with inverse-variance weighting was used in meta-analyses if any heterogeneity was detected, whereas a fixed-effect model was used if no heterogeneity was identified [36]. A value of *p* < 0.05 was considered statistically significant. We decided to examine publication bias with Egger’s test and funnel plots if the number of studies was 10 or above, since these analyses are underpowered otherwise. Six domains were evaluated: random sequence generation, allocation concealment, blinding of outcome assessor, completeness of outcome data reporting, selective outcome reporting, and other potential sources of bias.

## 3. Results

### 3.1. Study Characteristics

A search of the Medline (PubMed) database up to 12 February 2021 resulted in 50 relevant articles. Further search of other electronic databases yielded no additional articles (Figure 1).

No additional eligible studies were retrieved by hand searching bibliographies. Of these studies, 39 were excluded, according to our inclusion/exclusion criteria, as they evaluated other therapies than conization/LEEP or provided no adjuvant HPV vaccination. Eleven studies fulfilled the predefined inclusion criteria and were eligible for inclusion in the final meta-analysis [37,38,39,40,41,42,43,44,45,46,47]. In total, the studies comprised 21,310 patients: 4039 (19%) received adjuvant HPV vaccination while 17,271 (81%) surgery alone. The main characteristics of the studies included are summarized in Table 1.

All studies were published between 2012 and 2020. The study designs included: three prospective non-randomized studies [41,44,46], one randomized controlled trial [42], four retrospective studies [38,43,45,47], and three post-hoc pooled analyses of randomized clinical trials [37,39,40]. The women included in the studies were between the years of 15 and 89. The median follow-up time across the studies ranged from 2 to 5 years. HPV vaccination was administered after surgical treatment in nine studies [37,38,39,40,41,42,45,46,47], while either shortly before or after surgery in the other two studies [43,44]. The type of vaccine used was quadrivalent (against HPV 6/11/16/18 genotypes) in four studies [37,38,41,42] and bivalent (against HPV 16/18 genotypes) in two [39,40], whereas the other five studies [43,44,45,46,47] used both vaccines.

### 3.2. Risk of Bias

The risk of bias of included studies is reported in Figure 2.-Random sequence generation (selection bias): A total of 4 studies [37,39,40,42] gave an account of the generation of randomization sequence, thus we rated these as having low risk of bias for this item. All other studies were rated as having high risk of bias for this domain.-Allocation concealment (selection bias): A total of 4 studies [37,39,40,42] gave an account of concealing allocation to the intervention or control group, thus we rated these as having low risk of bias for this item. All other studies were rated as having high risk of bias for this domain.-Blinding of participants and personnel (performance bias): Only two studies [43,44] did not report actual procedures for blinding of participants and personnel, thus were considered at high risk of bias for this item. All other studies were rated as having low risk of bias for this domain.-Blinding of outcome assessment (detection bias): Only two studies [44,45] did not report actual procedures for blinding of outcome assessment, thus were considered at high risk of bias for this item. All other studies were rated as having low risk of bias for this domain.-Incomplete outcome data (attrition bias): All studies reported complete outcome data, thus we rated all them as having low risk of bias for this item.-Selective reporting (reporting bias): We assessed six studies [41,42,43,45,46,47] as having low risk of bias and three [38,40,44] as having high risk for this item. Two studies [37,39] did not report clear data on this specific item, thus the risk of bias was classified as unclear.

### 3.3. Effects of Interventions

#### 3.3.1. Primary Outcomes

##### CIN 2+ Recurrence

All included studies [37,38,39,40,41,42,43,44,45,46,47] evaluated the CIN 2+ recurrence, regardless of HPV types, within 6–60 months after treatment. Of the 21,310 women included in the pooled analysis (4039 in the vaccinated and 17,271 in the unvaccinated group), the CIN 2+ recurrence, regardless of HPV types, occurred in 1034 women (4.8%): 121 (3.0%) in the vaccinated and 913 (5.3%) in the unvaccinated cohort. Heterogeneity for this comparison was I^2^ 59% (95% CI 20.2–78.9%). The pooled estimated odds ratio (OR) was 0.35 (95% CI 0.21–0.56; *p* < 0.0001) (Figure 3).

Five studies [37,38,39,40,43], with a total of 2359 patients (1054 in the vaccinated and 1305 in the unvaccinated group), reported data on the CIN 2+ recurrence correlated with HPV 16/18, which occurred in 54 women (2.3%): 12 (1.1%) in the vaccinated and 42 (3.2%) in the unvaccinated cohort. Heterogeneity for this comparison was I^2^ 4% (95% CI 0–85.3%). The pooled estimated OR in this subgroup was 0.35 (95% CI 0.17–0.69; *p* = 0.003) (Figure 3).

Moreover, a sensitivity analysis was performed according to the study design dividing prospective and retrospective studies confirming a lower rate of CIN 2+ recurrence in the vaccinated compared to unvaccinated group. Heterogeneity for this comparison was I^2^ 65% (95% CI 0–88.1%). The pooled estimated OR for this subgroup in prospective trials was 0.36 (95% CI 0.14–0.91; *p* < 0.003) (Figure 4).

##### CIN 1+ Recurrence

Seven studies [37,38,39,40,41,42,45], with a total of 3375 patients (1609 in the vaccinated and 1766 in the unvaccinated group), evaluated the CIN 1+ recurrence, regardless of HPV types, within 6–48 months after surgical treatment. The CIN 1+ recurrence, regardless of HPV types, occurred in 287 women (8.4%): 102 (6.3%) in the vaccinated and 185 (10.5%) in the unvaccinated cohort. Heterogeneity for this comparison was I^2^ 65% (95% CI 21.3–84.4%). The pooled estimated OR was 0.51 (95% CI 0.31–0.83; *p* = 0.006) (Figure 5).

Four studies [37,38,39,40], with a total of 2240 patients (1004 in the vaccinated and 1236 in the unvaccinated), reported data on the CIN 1+ recurrence correlated with HPV 16/18, which occurred in 49 patients (2.2%): 11 (1.1%) in the vaccinated and 38 (3.1%) in the unvaccinated. Heterogeneity for this comparison was I^2^ 0% (95% CI 0–84.7%). The pooled estimated OR in this subgroup was 0.36 (95% CI 0.18–0.72; *p* = 0.004) (Figure 5).

Moreover, a sensitivity analysis was performed according to the study design dividing prospective and retrospective studies confirming a lower rate of CIN 1+ recurrence in the vaccinated compared to unvaccinated group. Heterogeneity for this comparison was I^2^ 0% (95% CI 0–90%). The pooled estimated OR for this subgroup in prospective trials was 0.20 (95% CI 0.07–0.54; *p* < 0.001) (Figure 6).

#### 3.3.2. Secondary Outcomes

##### HPV Persistence

Three studies [40,41,47] with a total of 955 patients (414 in the vaccinated and 541 in the unvaccinated) evaluated the risk of HPV persistence during follow-up reporting a no statistically significant increase in the incidence of HPV persistence. Heterogeneity for this comparison was I^2^ 0% (95% CI 0-89.6%). The pooled estimated OR was 0.84 (95% CI 0.61–1.15; *p* = 0.28) (Figure 7).

##### CIN 3 Recurrence

One study [37] reported the number of CIN 3 recurrences within 30 months after surgery. Of the 1066 women, 474 women were vaccinated and 592 were unvaccinated. There were 16 total cases (1.5%) of subsequent CIN 3 recurrence: three cases (0.6%) in the vaccinated group and 13 cases (2.2%) in the unvaccinated one (OR 0.28; 95% CI 0.08–1.00; *p* = 0.05). No data about CIN 3 recurrences correlated with HPV 16–18 were reported.

Persistence of Abnormal Cervical Cytology

One study [37] evaluated persistence of positive pap smear at first clinical evaluation after surgery. Of the 178 women, 89 women were vaccinated and 89 were not. The persistence of abnormal cervical cytology was reported in 10.1% (9/89) of unvaccinated patients and none (0%) of vaccinated ones (OR 0.05; 95% CI 0.01–0.83; *p* = 0.04).

## 4. Discussion

The beneficial role of HPV vaccines in protecting against cervical cancer by lowering the risk of getting infected with HPV has been fairly established, especially when they are given before initiating sexual activity [6,9]. Conversely, it is still controversial whether they are useful even after HPV infection has already occurred. Accumulating data highlighted that potentially HPV vaccines have a role in the adjuvant setting [18,19,20]; however, the level A evidence is still lacking.

Several studies suggested that HPV vaccination has a statistically significant impact in reducing post-surgical recurrent disease both in women and men exposed to previous HPV infection [37,48,49]. In 2012, Joura et al. reported that HPV vaccination among women who had surgical treatment for HPV-related disease significantly reduced the incidence of subsequent CIN, vaginal intraepithelial neoplasia (VaIN), vulvar intraepithelial neoplasia (VIN) and genital warts [37]. In 2014, Coskuner et al. confirmed also in men the significant impact of adjuvant HPV vaccination in reducing the risk of recurrence of genital warts after initial surgical treatment [48]. In 2021, Ghelardi et al. added interesting insights to the current knowledge about the mechanism of post-surgical adjuvant HPV vaccination by documenting the effectiveness of HPV vaccine in preventing recurrent disease after surgical treatment for vulvar high-grade squamous intraepithelial lesions (HSIL) [49].

The present meta-analysis demonstrates that HPV vaccination is effective as an adjunct to surgery in preventing the risk of relapse for cervical dysplasia. The overall risk reduction of having a new or persistent CIN 2+ after surgery was 65%. The pooled estimated OR was 0.35 (95% CI 0.21–0.56; *p* < 0.0001). This clinical benefit is equally evident when restricting the analysis to the two most oncogenic HPV types (HPV 16 and 18), that are included in the vaccine. These results may have significant clinical implications considering that the risk of relapse after the surgical treatment is relatively high, ranging from 9% to 14% [14].

In this intriguing scenario, supporting the role of adjuvant vaccination, some relevant points need to be addressed. In particular, the optimal timing for vaccination remains to be clarified. Vaccination was administered either shortly before or after the surgical treatment, and the precise timing varied across the studies. In particular, in 7 studies [37,38,39,41,42,45,46] the first dose of vaccination was administered within the first month after surgery. Two studies [43,44] reported that the first dose of vaccination was administered between 3 months before surgery and 12 months after surgery, while the other 2 studies [40,47] did not specify the exact timing of the first dose (Table 1). The lack of standardization of the timing for HPV vaccination prevented from making any comparisons of CIN recurrence between women who received vaccination before and after surgery.

Therefore, future prospective studies are required to properly standardize the time of administration, which would probably be appropriate within 30 days from surgery. Furthermore, more data are required to clarify whether the post-conization HPV vaccination is able to ensure a benefit to the overall population, or only to a restricted group of women such as those with documented HPV persistent infection. Awaiting more consolidated data on these specific points, it should be acknowledged that our meta-analysis has several strengths and limitations. Strengths include the following: (a) a comprehensive evaluation of all currently available data providing a large sample size (21,310 patients); (b) the prospective nature of four studies, with one true randomized controlled trial investigating the utility of adjuvant HPV vaccination after surgical excision for CIN 2+ compared with placebo or surgical procedure alone; (c) the quality of the methodology used along with the strict inclusion criteria, in order to specifically answer the proposed question. There are also several limitations of this study: (a) different inclusion criteria and methodologies (e.g., type of vaccine, time of vaccination, and follow-up) were used, resulting in significant heterogeneity between studies; (b) since the published evidence on this topic is lacking, both randomized and non-randomized studies were included in the final analysis; (c) one study [44] accounted for nearly 80% of patients analyzed, potentially leading to selection and information.

All studies except one are concordant in suggesting the beneficial role of HPV vaccination [26,27,28,29,30,31,32,33,34,35,36]. It should be pointed out that, unlike all the other studies included in this meta-analysis, Hildesheim et al. [40] was the only one not reporting a significant risk reduction of CIN recurrence for adjuvant vaccination. However, as it was a subgroup analysis of patients who were not separately randomized, the results reported by Hildesheim et al. may be affected by unavoidable bias.

As a consequence of the growing attention on the role of post-LEEP vaccination, three meta-analyses focused on this specific point have been recently published. Interestingly, these manuscripts showed superimposable findings, all in accordance with our data (Bartels et al. (RR 0.51; 95% CI, 0.35–0.74) [18], Lichter et al. (RR 0.36; 95% CI, 0.23–0.55) [19], and Jentschke et al. (RR 0.41; 95% CI, 0.27–0.64) [20]). However, when compared with the most recent one by Jentschke et al. our meta-analysis added a new multicenter, retrospective study by Bogani et al. [47], with 300 patients selected for the propensity-score matching analysis and a median follow-up of 5 years. Unlike the meta-analysis by Lichter et al. [19], which included also the study by Grzes’ et al. [50], our meta-analysis excluded this study, as there was no full text available in English language.

Besides the above-mentioned limitations, focusing on clinical practice, by pooling the currently available data on this topic, it can be made an initial and interesting association between the reduction of CIN recurrence and the administration of perioperative HPV vaccination. However, this is still not enough to allow the application of HPV vaccination in this setting as an evidence-based routine practice. Even if some physicians may already rely on these early results and recommend vaccination after surgery for cervical dysplasia, the costs of vaccination are still not covered by healthcare systems. Currently, there are three randomized controlled trials ongoing on clinicaltrials.gov: the NOVEL trial (NCT03979014; not yet recruiting), the VENUS study (NCT04171505; recruiting), and the COVENANT trial (NCT03284866; active, not recruiting). Results from these studies will probably provide new insights on this emerging topic.

## 5. Conclusions

The present meta-analysis of eleven studies demonstrates that using prophylactic HPV vaccination as an adjunct to surgical excision for CIN 2 or greater reduces the risk of recurrent disease (OR 0.36; 95% CI 0.22–0.57; *p* < 0.0001). Further large-scale, randomized controlled studies are required to confirm these findings and drive adjuvant HPV vaccine into routine clinical practice. Moreover, the optimal timing of vaccination remains to be clarified.

## Figures and Tables

**Figure 1 vaccines-09-00410-f001:**
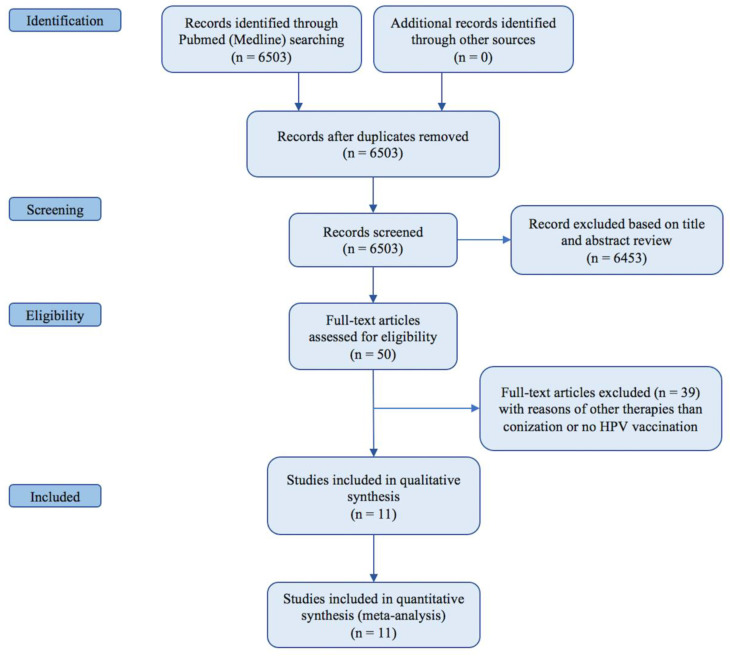
PRISMA flow chart.

**Figure 2 vaccines-09-00410-f002:**
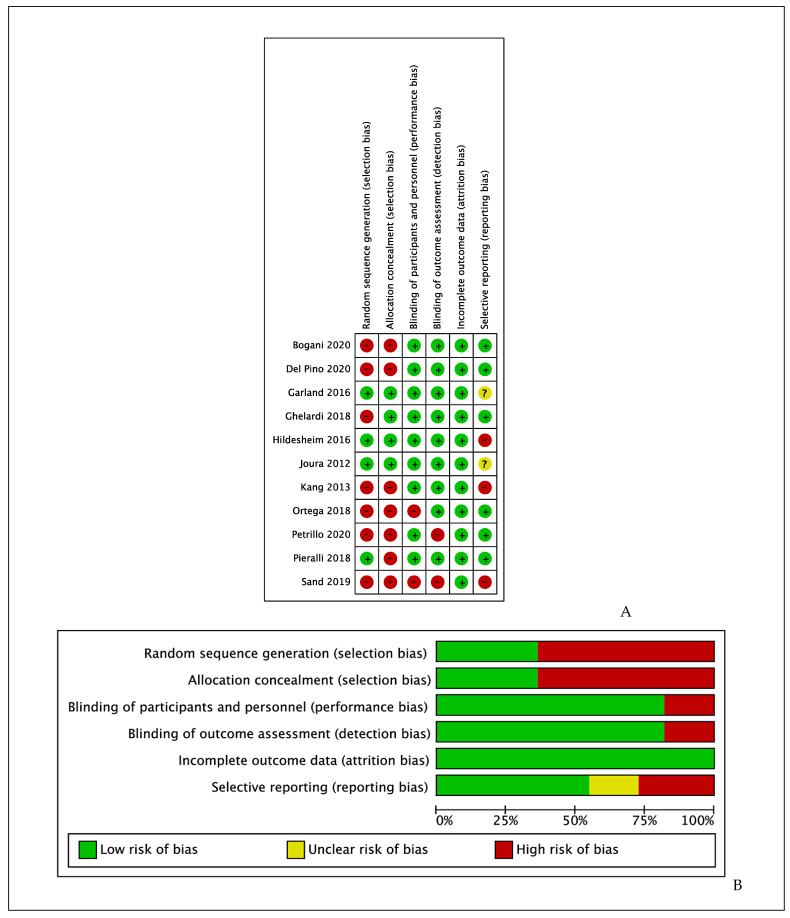
(**A**) Risk of bias summary: authors’ judgments about each risk of bias item for each included study. (**B**) Risk of bias graph: authors’ judgments about each risk of bias item presented as percentages for all included studies.

**Figure 3 vaccines-09-00410-f003:**
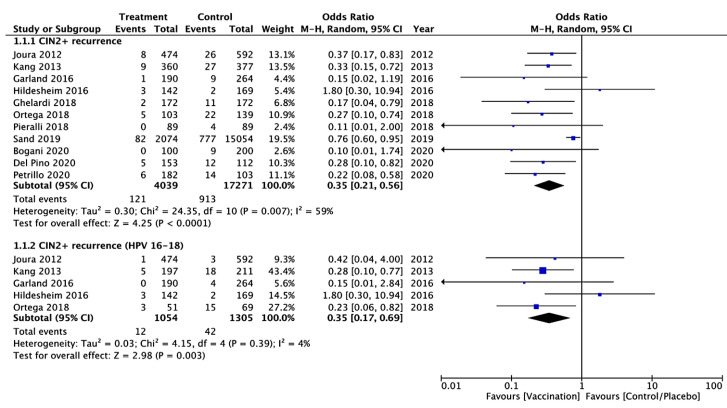
Forest plot of comparison: CIN 2+ recurrence regardless of HPV types and CIN2+ recurrence correlated with HPV 16/18.

**Figure 4 vaccines-09-00410-f004:**
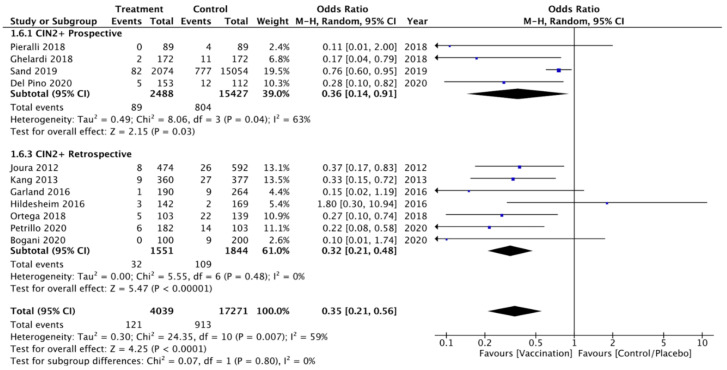
Forest plot of comparison: sensitivity analysis according to the study design for CIN 2+ recurrence.

**Figure 5 vaccines-09-00410-f005:**
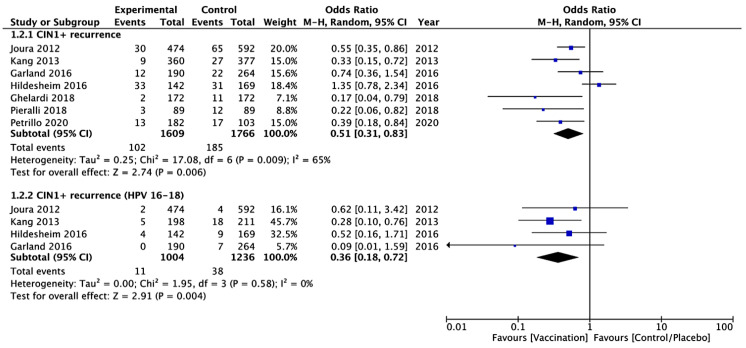
Forest plot of comparison: CIN 1+ recurrence regardless of HPV types and CIN 1+ recurrence correlated with HPV 16/18.

**Figure 6 vaccines-09-00410-f006:**
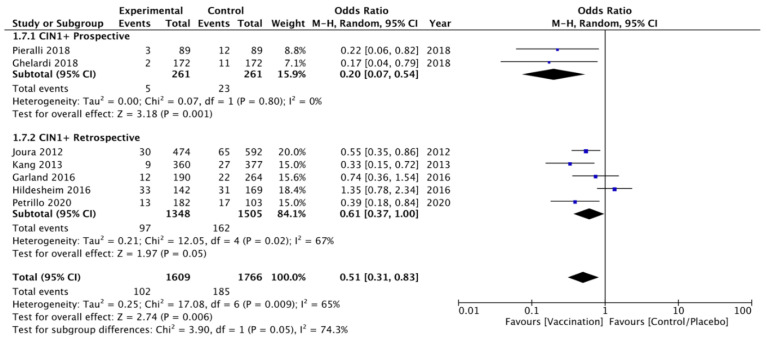
Forest plot of comparison: sensitivity analysis according to the study design for CIN 1+ recurrence.

**Figure 7 vaccines-09-00410-f007:**
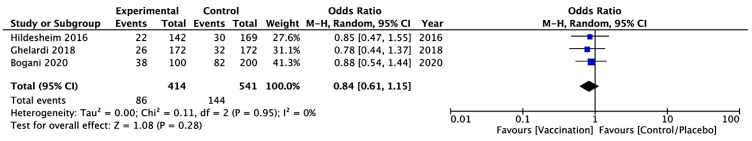
Forest plot of comparison: HPV persistence.

**Table 1 vaccines-09-00410-t001:** Summary of studies included in the systematic review and meta-analysis.

Study, Year	Study Design	N. of Patients Age (Years)	Primary Endpoint	HPV Vaccine Type and Time of Vaccination	Surgical Treatment
Joura et al., 2012 [37]	Post-hoc-pooled analysis of 2 RCT (FUTURE I and II) Follow-up 2.5 years (median)	1066 15–26	CIN 2+ (HPV-type independent)	Quadrivalent at day 1, month 2, and month 6 after surgery	LEEP (84.7%), cervical conization (12.5%), cryotherapy (0.7%), and other n.a. (2.1%)
			CIN 2+ (HPV 16, 18)		
			CIN 1+ (HPV-type independent)		
			CIN 1+ (HPV 16, 18)		
			CIN 3 (HPV-type independent)		
Kang et al., 2013 [38]	Retrospective case-control Follow-up 3.5 years (median)	737 20–45	CIN 2+ (HPV-type independent)	Quadrivalent at week 1, month 2, and month 6 after surgery	LEEP
			CIN 2+ (HPV 16, 18)		
			CIN 1+ (HPV-type independent)		
			CIN 1+ (HPV 16, 18)		
Garland et al., 2016 [39]	Post-hoc analysis of a RCT (PATRICIA) Follow-up 4 years	454 15–25	CIN 2+ (HPV-type independent)	Bivalent at months 0, 1, and 6 after surgery	LEEP
			CIN 2+ (HPV 16, 18)		
			CIN 1+ (HPV-type independent)		
			CIN 1+ (HPV 16, 18)		
Hildesheim et al., 2016 [40]	Subgroup analysis of a RCT Follow-up 27.3 mo (median)	311 18–25	CIN 2+ (HPV-type independent)	Bivalent, 3 doses over 6 months after surgery	LEEP
			CIN 2+ (HPV 16, 18)		
			CIN 1+ (HPV-type independent)		
			CIN 1+ (HPV 16, 18)		
Ghelardi et al., 2018 [41]	Prospective case-control (SPERANZA project) Follow-up 4 years	344 18–45	CIN 2+ (HPV-type independent)	Quadrivalent at day 30, month 2, and month 6 after surgery	LEEP
			CIN 1+ (HPV-type independent)		
Pieralli et al., 2018 [42]	RCT Follow-up 3 years	178 <45	CIN 2+ (HPV-type independent)	Quadrivalent at months 0, 2 and 6 after surgery	Conization (83%), other n.a. (17%)
			CIN 1+ (HPV-type independent)		
Ortega-Quinonero et al., 2019 [43]	Retrospective Follow-up 2 years	242 18–65	CIN 2+ (HPV-type independent)	Bi-/Quadrivalent, first dose 0-1 months before or 0-1 months after surgery, other 2 doses over 6 months	LEEP
			CIN 2+ (HPV 16, 18)		
Sand et al., 2020 [44]	Prospective cohort	17128 17–51	CIN 2+ (HPV-type independent)	Bi-/Quadrivalent, first dose 0-3 months before or 0-12 months after surgery	Conization
Petrillo et al., 2020 [45]	Retrospective Follow-up 2 years	285 32–47	CIN 2+ (HPV-type independent)	Bi-/Quadrivalent, first dose 0-1 months after surgery	LEEP
			CIN 1+ (HPV-type independent)		
Del Pino et al., 2020 [46]	Prospective Follow up 22.4 mo median	265 26–64	CIN 2+ (HPV-type independent)	Bivalent at 0, 1 and 6 months after surgery Quadrivalent at 0, 2 and 6 months after surgery	Conization
Bogani et al., 2020 [47]	Retrospective, multicenter Follow-up 5 years	300 18–89	CIN 2+ (HPV-type independent)	Bi-/Quadrivalent	LEEP

CI, confidence interval; CIN, cervical intraepithelial neoplasia; HPV, human papillomavirus; LEEP, loop electrosurgical excision procedure; RCT, randomized controlled trial; RR, relative risk.

## Data Availability

Not applicable.

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
