# Peer review of "Adjuvant HPV Vaccination to Prevent Recurrent Cervical Dysplasia after Surgical Treatment: A Meta-Analysis"

_vaccines, 2021, doi:10.3390/vaccines9050410_

Round 1

Reviewer 1 Report

This paper conducted a systematic literature review and meta-analysis to discuss evidence supporting the efficacy of adjuvant human papillomavirus (HPV) vaccination in reducing the risk of recurrent cervical intraepithelial neoplasia (CIN) 1+ after surgical treatment. The results indicate that HPV vaccination is associated with a reduced risk of recurrent CIN 1+ and CIN 2+ after surgical treatment, which drives adjuvant HPV vaccine into routine clinical practice. 

Major comments:

  1. Why is “Allocation concealment (selection bias)” [lines 181-183] identical to “Random sequence generation (selection bias)” [lines 177-180]? Please explain.
  2. Lines 202 to 207:
    • How come the CIN2+ occurred in 1,034 women but re-occurred in 2,359 women? Please explain.
    • Why did not mention in result that the CIN2+ recurrence in all women regardless of HPV types.
  3. Lines 215 – 216:
    • How come the CIN 1+ occurred in 287 women but re-occurred in 2,240 women? Please explain.
    • Why did not mention in result that the CIN1+ recurrence in all women regardless of HPV types.
  4. Vaccine timing: Authors claimed that “the timing of vaccination has not demonstrated to have any significant influence on the recurrence rate (line 260-262)” but did not show the data for the claim. It is interesting to see the comparison of recurrence in women received vaccination before and after surgery. Or please explain why authors did not do the analysis.

 Minor comments:

  • Line 20-21: change “4039 (19%) received adjuvant HPV vaccination while 17271 (81%) surgery alone” to “4039 (19%) received peri-operational adjuvant HPV vaccination while 17271 (81%) surgery received surgery alone”.
  • Line 33: change “Human papillomavirus (HPV) infections are responsible for almost all cases of cervical cancer” to “Human papillomavirus (HPV) infections are responsible for majority of cervical cancer cases.” Since there is still a proportion of true HPV negative cervical cancers [TJALMA WAA, PMID: 31110650]
  • Line 73: change “there is currently not sufficient evidence” to “there is currently no sufficient evidence”.
  • Line 147: is 12/02/2021 the correct date? Should it be 12/02/2020?
  • Line 168: change “while either shortly before or after in two studies.” to “while either shortly before or after surgery in the other two studies.”
  • Table 1:
    • Please change “years” to “age (years)”.
    • Why are some texts bold and others are not? Please clarify.
    • Why were HPV-independent cases included in the study? Or these were typos and should be HPV-type independent?
  • Line 195: please add a space between “[29,31,35]” and “as”.
  • Line 217 - 218: please de-gap between “infection” [line 219] and “on” [line 220].
  • Line 227 - 229: please de-gap between “un-vaccinated” [line 227] and “evaluated” [line 229].
  • Please unify CIN 1+ or CIN1+ throughout the text. The same apply to CIN2+ and CIN3.
  • Line 237: please change “subsequent CIN 3” to “subsequent CIN 3 recurrence”.
  • Line 261: please remove “a” between “any” and “significant”.

Reviewer 2 Report

With interest, I read the review “Adjuvant HPV vaccination to prevent recurrent cervical dysplasia after surgical treatment: A meta-analysis” by Di Donato and coll. There are some concerns that must be addressed before accepting it for publication in Vaccines.

Introduction

The first sentence needs reference.

Again, some context data on HPV-attributable cancer-mortality could be added to give more strength to the findings of this SR/MA.

Methods

Time and language limits have been set for the literature search? Please specify.

  1. 84. It should be reported the complete search strategy used, at least for the principal database surfed (e.g., PubMed).

Exclusion criteria 4-7 must be better clarified.

Is there any reference for the quality score used? If not, please use an evidence-based score to allow comparability.

Statistical analysis

I hardly understand “A random-effect model was used at meta-analysis if any heterogeneity was detected, whereas a fixed-effect model was used if no heterogeneity was identified.” The choice of the model should rely on correlation between the individual-specific effects and independent variables, and not depends on the presence/absence of heterogeneity. Again, I suggest to add (i) subgroup analyses (e.g., with stratification according to study design or quality assessment); (ii) sensitivity analyses.

Results

Fig 1 should follow the PRISMA format.

Discussion

More comparison with other existing similar studies is needed (for example, with studies conducted in HPV vaccination amongst men after anal or penile pre-cancerous lesion/cancer and so on)

Round 2

Reviewer 2 Report

Authors fully addressed my previous comments and suggestions. I can advise that the manuscript is now acceptable for publication.